# Looking for the Holy Grail—Drug Candidates for Glioblastoma Multiforme Chemotherapy

**DOI:** 10.3390/biomedicines10051001

**Published:** 2022-04-26

**Authors:** Beata Pająk

**Affiliations:** Independent Laboratory of Genetics and Molecular Biology, Kaczkowski Military Institute of Hygiene and Epidemiology, Kozielska 4, 01-163 Warsaw, Poland; bepaj@wp.pl

**Keywords:** glioblastoma multiforme (GBM), drug candidates, clinical trials, glycolysis inhibitors, kinases inhibitors, immunomodulatory action

## Abstract

Glioblastoma multiforme (GBM) is the deadliest and the most heterogeneous brain cancer. The median survival time of GBM patients is approximately 8 to 15 months after initial diagnosis. GBM development is determined by numerous signaling pathways and is considered one of the most challenging and complicated-to-treat cancer types. Standard GBM therapy consist of surgery followed by radiotherapy or chemotherapy, and combined treatment. Current standard of care (SOC) does not offer a significant chance for GBM patients to combat cancer, and the selection of available drugs is limited. For almost 20 years, there has been only one drug, Temozolomide (TMZ), approved as a first-line GBM treatment. Due to the limited efficacy of TMZ and the high rate of resistant patients, the implementation of new chemotherapeutics is highly desired. However, due to the unique properties of GBM, many challenges still need to be overcome before reaching a ‘breakthrough’. This review article describes the most recent compounds introduced into clinical trials as drug candidates for GBM chemotherapy.

## 1. Introduction

### 1.1. Glioblastoma Multiforme—An Overview

Glioblastoma multiforme (GBM) is one of the most complex, deadly, and treatment-resistant cancers. The five-year survival rate for glioblastoma patients is about 5 percent. The average length of survival is estimated to be only 12 to 18 months with treatment and 3 to 4 months without treatment [1]. The first-line SOC treatment is based on complete surgical resection, followed by concomitant administration of radio-and chemotherapy and adjuvant chemotherapy (Stupp protocol) [2]. According to statistics, age, Karnofsky Performance Scale (KPS), and methylguanine-DNA methyltransferase (MGMT) status are the major predictors of survival [3,4]. Unfortunately, in most cases, recurrence occurs within 8–10 months of initial resection [5]. The median survival for patients with recurrent GBM is around 9 months [5]. According to recent studies by Sacko et al. [2], repeat surgery was beneficial in more than 30% of the patients who underwent repeat resection. However, it should be underlined that repeat surgery is only possible in patients with a high KPS score, which is not often the case for advanced GBM patients. Nevertheless, regardless of the applied treatment protocol, the prognosis for GBM patients is extremely poor, and novel, more effective therapeutic strategies are highly desired.

### 1.2. Standard Therapy—Chemotherapy

Chemotherapy is one of the main therapeutic procedures in GBM patients. Usually, chemotherapy is introduced two to four weeks after surgery, combined with or shortly after radiotherapy (RT). If the tumor cannot be surgically removed, chemotherapy is a primary treatment [6]. 

Standard chemotherapy consists of Temozolomide (TMZ) (75 mg/m^2^ daily) during radiotherapy, followed by a further six cycles of TMZ (150–200 mg/m^2^ on days 1–5 every 28 days) [6]. It should be underlined that TMZ was approved in 2005 [7] and is still the only available first-line drug in GBM treatment. In recent decades, several promising novel molecules have been investigated, but most of them failed clinical trials due to systemic toxicity and poor drug delivery to the brain. The major obstacle in the treatment of GBM is the blood–brain barrier (BBB), which limits the penetration of relatively high molecular weight compounds to the tumor site. Thus, regardless of current achievements in advanced anticancer therapies, the progress of GBM therapy has been dismal and almost unchanged for almost two decades [8].

TMZ is an imidazotetrazinone derivative. TMZ is hydrolyzed inside the cells into active metabolite 5-(3-methyl-1-triazen-1-yl)imidazole-4-carboxamide, which further degrades into methyl diazonium ion [9]. Methyl diazonium ion transfers a methyl group at the N-7 and O-6 positions of guanine and the O-3 position of adenine in the DNA structure. The methylation of the O-6 position of guanine is the most cytotoxic, which results in a ‘mismatch’ incorporation of thymidine instead of cytosine [10]. Consequently, proper DNA replication is altered, leading to cell death induction. On the other hand, thymidine incorporation is recognized by the mismatch repair (MMR) enzymes, such as o-6-methylguanine-DNA methyltransferase (MGMT), which repair the guanine methylation, leading to the escape of GBM cells from TMZ-dependent apoptosis [11]. It is estimated that at least 50% of GBM patients do not respond to TMZ, due to the overexpression of the MGMT enzyme [12]. Regardless of MGMT status, other molecular mechanisms of drug resistance determining the limited response to TMZ treatment are also reported, such as ferroptosis [13] and the upregulation of MGMT-independent DNA repair mechanisms, such as Base Excision Repair (BER) [14], Multidrug Resistance Protein 1/P-glycoprotein (MDR1/Pgp) activity [15], and the presence of cancer stem cells [CSC] [16]. 

Apart from TMZ’s insufficient clinical efficacy, the severe limitation of TMZ is its toxicity, including nausea, vomiting, fatigue, and myelosuppression, especially thrombocytopenia and neutropenia. Additionally, serious hematologic adverse events, including myelodysplastic syndrome and aplastic anemia, have been reported [17].

Unfortunately, even when glioblastoma is discovered and treated aggressively, it almost always recurs. This very high recurrence rate is why there are so few long-term survivors of the disease. Even when the tumor appears to have been eliminated, the median time to recurrence is 9.5 months, with an overall survival of approximately 30 weeks [18]. There is no standard treatment for recurrent GBM; however, alkylating chemotherapy is commonly used. The second line of GBM chemotherapy includes a procarbazine/lomustine/vincristine (PCV) or Bevacizumab/irinotecan (BI) scheme [19]. According to studies summarized by Carvalho et al. [19], the overall response rate in the BI group was 66%, while it was only 11% in the PCV group. The median progression-free survival (PFS) was 5 and 3 months, respectively. The median overall survival (OS) was 9 months in the BI group and 5 months in the PCV group. More importantly, the toxicity was also significantly lower in the BI group [19]. Regardless of the described combinations, various chemotherapeutics are also applied in different regimens. For example, lomustine (60 mg/m^2^ on days 8–21/56), vincristine (1.4 mg/m^2^ on days 8 and 29/56), cisplatin (60–100 mg/m^2^ once every 3–4 weeks), etoposide (50 mg daily), and procarbazine (110 mg/m^2^ day 1/56) [20] are used in GBM treatment. Unfortunately, the majority of them have severe side effects, significantly lowering the quality of life of GBM patients. 

In 1997, the FDA-approved Gliadel wafers for recurrent GBM therapy. A few years later, in 2003, Gladel was accepted for primary high-grade glioma (grade III and IV according to the World Health Organization (WHO) classification) treatment. Currently, Carmustine is used in primary (GBM, medulloblastoma, astrocytoma, ependymoma) and metastatic brain tumor therapy. Other cancers treated with Carmustine include multiple myeloma; Hodgkin’s and non-Hodgkin’s lymphomas; and melanoma, lung, and colon cancer [21]. Gliadel—Carmustine-impregnated wafers—are a cell-cycle nonspecific alkylating agent that causes cross-links in DNA and RNA, leading to the inhibition of DNA synthesis, RNA production, and translation. Carmustine can also bind to and modify glutathione reductase [21]. Carmustine wafer formulation, the Gliadel wafer, is implanted into a surgical cavity after removing a brain tumor to release Carmustine. Carmustine wafers (Gliadel^®^ Wafer) significantly improved the safety and the therapeutic efficacy of Carmustine by its controlled release from biodegradable polymer wafers to the local GBM tumor environment. Thus, Gliadel is currently approved for use in recurrent GBM patients and newly diagnosed patients with high-grade glioma as an adjuvant to surgery with/without RT. Furthermore, according to a meta-analysis of more than 5800 GBM patients treated with Carmustine [21], both newly diagnosed and recurrent GBM patients who received Carmustine (as implantation in a resection cavity or intravenous administration) treatment showed a prolonged OS. Moreover, GBM patients treated with combined Carmustine and TMZ therapy also had longer OS than those receiving TMZ alone [21]. Based on the available data, it can be concluded that Carmustine implantation in the resection cavity provides survival benefits for GBM patients. It may be a promising supplement to the standard therapeutic protocol by offering a bridge between surgical resection and the onset of TMZ therapy.

In May 2009, the US FDA granted accelerated approval to a monoclonal antibody against vascular endothelial growth factor A (VEGF)—an important molecule promoting endothelial cell proliferation, new vessel formation, and subsequent rapid tumor growth [22] in recurrent GBM patients. Approval of Bevacizumab raised great hopes for a more effective treatment outcome in GBM. Initially, Bevacizumab’s early phase 2 data indicated improved PFS, although no OS benefit was seen [23,24]. A subsequent randomized phase 3 trial demonstrated that Bevacizumab combined with lomustine improved PFS compared with lomustine monotherapy, but again without significant improvement in OS [25]. Typically, the combination of Bevacizumab with chemotherapy is introduced after failure with Bevacizumab monotherapy. A wide range of drugs, including lomustine, Carmustine, and TMZ, have been tested [26,27,28]. Unfortunately, no data demonstrate a survival benefit with any of these regimens. Moreover, the last registered clinical trial of Gliadel was dedicated to newly diagnosed GBM patients treated with Gliadel, followed by radiotherapy (RT), TMZ, and Bevacizumab, followed further by Bevacizumab and TMZ post-radiation. However, according to the Clinical Trials.gov database, this study (NCT01186406) was terminated due to observed toxicity. The lack of OS benefit with single-agent Bevacizumab limits its approval in Europe. Currently, the results show that Bevacizumab can improve quality of life with decreased corticosteroid use and is thus sometimes applied to patients with later recurrences [28].

## 2. GBM Drug Candidates in Clinical Trials

As summarized above, the chemotherapy options for GBM therapy are minimal. Therefore, intensive efforts aim to model and synthesize new molecules with anticancer potential. Moreover, drugs registered in other indications are being investigated as potential components of combination therapy in GBM therapy. Regardless of the significant progress in anticancer drug development, GBM remains one of the most challenging targets. Numerous preclinical in vitro studies showed cytotoxic efficacy of various compounds against GBM cells. However, the transition of preclinical results to clinical efficacy is extremely rare. Thus, when discussing possible new drugs for GBM, we should focus on current clinical trials. Table 1 summarizes the latest clinical trials (from 2016 to the present) for GBM chemotherapy (monotherapy or combination) or as an adjuvant for radiotherapy (RT). 

As shown in Table 1, almost all tested drug candidates are at the initial stages of their clinical development (phase 1 or 2). The only phase-3-advanced drug is Nivolumab in a combination treatment with SOC or Bevacizumab. Trans sodium crocetinate (NCT03393000), which increases the oxygenation of hypoxic tissues, leading to cell death, was terminated in its phase 3 trial due to the business decisions of the sponsor. 

Among the tested drugs, most of the small molecule compounds could be administrated orally (PO) or intravenously (IV). In the case of monoclonal antibodies (mAb), the main obstacle is the effective penetration of the drug into the brain tumor tissue. Thus, mAbs are delivered by intramuscular (IM) injection followed by electroporation (EP) (NCT03491683). Numerous tested drugs are combined with the current SOC, TMZ +/− RT. It should be underlined that even in the presence of MGMT methylation, the overall survival of TMZ-treated patients is only a few months longer than in the MGMT-unmethylated group. Thus, there is an unmet medical need for new chemotherapeutics that can replace TMZ as a first-line treatment with significant efficacy and a high safety profile for patients.

### 2.1. Drug Candidates Targeting GBM Cancer Cell Metabolism

Rapid tumor growth quickly outstrips the capacity of oxygen diffusion. To overcome this problem, tumors form new blood vessels via angiogenesis. Unfortunately, the unstable tumor neovasculature causes local hypoxia and subsequent lactic acidosis. In response to such unfavorable conditions, cancer cells use anaerobic metabolism as the major pathway of adenosine triphosphate (ATP) synthesis [29]. It was shown that aerobic glycolysis promotes tumor growth by increasing cell biomass. Several cell proliferation signaling pathways also regulate cell metabolism via incorporating nutrients into tumor biomass. Thus, glycolysis could be maintained as the primary pathway for glucose metabolism, even in the presence of oxygen [30]. In response to hypoxic conditions, several protooncogenes (e.g., c-Myc), signaling pathways (e.g., phosphatidylinositol-3-kinase (PI3K/Akt)), and specific transcription factors (e.g., hypoxia-inducible factor 1 alpha, HIF-1a) are activated [30]. The HIF-1a transcription factor is a major driver of cancer cell metabolism reprogramming [31]. It regulates the expression of glucose transporters and glycolytic enzyme genes. On the other hand, HIF-1 augments mitochondrial respiration [32] and regulates the balance between oxygen consumption and ATP and toxic reactive oxygen species (ROS) synthesis [33]. Reprogramming of tumor cell metabolism from oxidative phosphorylation to aerobic glycolysis is a major strategy for determining cancer cell survival under hypoxic conditions. While not all types of cancer rely only on anaerobic glucose utilization, the most aggressive tumors, such as pancreatic ductal adenocarcinoma (PDAC), and glioblastoma multiforme (GBM), entirely depend on glycolysis [34]. 

In our opinion, in the case of GBM, the old and simple concept of targeting cancer cells’ metabolism could be an effective strategy to implement new drugs for GBM therapy. Since the 1950s, a synthetic analog of glucose—2-deoxy-D-glucose (2-DG)—has been intensively tested in various cancer models, including GBM [34]. However, despite numerous preclinical and clinical studies, the use of 2-DG in GBM treatment has not been approved. The main obstacle is the rapid metabolism and short half-life of 2-DG (not more than 1 h) [35]. Moreover, 2-DG has to be used at relatively high concentrations (≥5 mmol/L) to compete with blood glucose [36]. 

Dichloroacetate (DCA) is similar to 2-DG, a glycolysis inhibitor tested in a phase 2A clinical trial (NCT05120184). It is noteworthy that DCA has been approved as an orphan drug for congenital lactic acidosis treatment [37]. DCA is a potent pyruvate dehydrogenase kinase (PDK) inhibitor. This mitochondrial enzyme is activated in various cancers, including GBM. It results in the selective inhibition of pyruvate dehydrogenase, a complex of enzymes that converts cytosolic pyruvate to mitochondrial acetyl-CoA, the substrate for Krebs’ cycle [38]. Inhibition of PDK with DCA shifts cancer cell metabolism from glycolysis to mitochondrial oxidation. Moreover, DCA reverses the suppression of mitochondria-dependent apoptosis by increasing the production of diffusible Krebs’ cycle intermediates and mitochondria-derived reactive oxygen species, following activation of p53 and HIF-1α inhibition [38]. Importantly, DCA is a small molecule of 150 Da, which partially explains its high bioavailability and ability to penetrate the BBB [39]. Due to these features, DCA is considered a drug candidate for GBM therapy. Importantly, due to the approved use of DCA in congenital lactic acidosis therapy, long-term safety studies are available [37]. DCA is generally well-tolerated, and no evidence of severe hematologic, hepatic, renal, or cardiac toxicity has been reported [40]. Common gastrointestinal side effects occur in most patients treated with DCA [41]. The best-known limitation of DCA administration, observed in preclinical and clinical studies, is peripheral neuropathy, which could be partially reduced by intravenous instead of oral administration [42]. Numerous preclinical studies also suggest positive effects of DCA co-administered with compounds currently used to treat other diseases but showing anticancer properties in several cancer models [43]. Skeberdytė et al. [44] found that administration of DCA and the antibiotic Salinomycin exerted a synergistic cytotoxic effect by inhibiting the expression of proteins related to multidrug resistance in colorectal cancer cell lines. On the other hand, a combined administration of recombinant arginase and DCA had antiproliferative effects in triple-negative breast cancer due to the activation of p53 and cell cycle arrest [45]. The subsequent study performed in cervical cancer cells showed that COX2 inhibition by celecoxib makes cervical cancer cells sensitive to DCA treatment [46]. 

In the head and neck cancer model, the combination of DCA with propranolol also appeared to be an efficient cell death inducer [47]. A positive effect of DCA coadministration with metformin, a drug widely used to treat diabetes, was demonstrated in a preclinical model of glioma [48], as well as in a low metastatic variant of Lewis lung carcinoma (LLC) [49].

In 2010, Michelakis et al. [40] described the results of oral DCA administration at a dose of 15.5 mg/kg twice a day in five patients with GBM. Three of these patients had recurrent GBM with disease progression after SOC. The initial dosage did not cause significant toxicity. Only peripheral neuropathy was noted but was reversed upon a dose decrease to 6.25 mg/kg of DCA twice a day. After 18 months of DCA treatment, three of the patients showed evidence of tumor regression, while four of five patients were clinically stable [40]. The authors also verified the presence of apoptotic cells, ROS levels, and HIF-1α expression in three GBM tissues from pre-DCA and post-DCA biopsies. They found that chronic DCA administration resulted in decreased proliferation and increased apoptosis, mitochondria depolarization, increased ROS, and decreased HIF-1α expression and activity, resulting in the significant inhibition of tumor vascularity compared to the pre-DCA GBM level. This mechanistic study in patients with GBM, although small, established the fact that after oral therapy, DCA can be further considered as a novel approach in brain tumor therapy. 

The currently registered clinical trial with DCA (NCT05120184) is a multicenter, open-label 2A trial of oral DCA in 40 surgical patients with recurrent GBM who have clinically indicated plans for debulking surgery. Patients will be genotyped to established safe dosing regimens and randomized to receive DCA (*n* = 20) or no DCA (*n* = 20) for one week before surgery. The study’s completion date is estimated to be June 2025. 

The next drug candidate targeting GBM cell metabolism is anhydrous enol-oxaloacetate (AEO), registered for a phase 2 clinical trial (NCT04450160) in newly diagnosed GBM patients who will receive SOC treatment along with AEO treatment. In July 2020, the FDA granted Fast Track designation for AEO development in primary GBM therapy. 

In the body, AEO is metabolized to oxaloacetate (OAA). Oxaloacetate is a key anaplerotic substrate required to maintain TCA cycle flux. Ruban et al. [50] showed that glutamate scavengers, including OAA, can inhibit tumor growth in glioma animal models. This ability stems from the fact that excess glutamine in the peritumoral space of gliomas plays a crucial role in glioma invasiveness. Glutamine released via the glutamine-cysteine exchanger kills neighboring neurons and allows the tumor to occupy an ever-increasing space within the brain [50]. Thus, glutamine reduction via OAA treatment in mice and rats with implanted glioma cells resulted in a significant tumor growth reduction and prolonged survival. These effects were also improved when OAA was combined with TMZ [50]. The anticancer potential of AEO and OAA has also been confirmed by Ijare et al. [51], who reported altered bioenergetics of GBM cells, specifically glucose oxidation. Apart from regulating glutamine metabolism, in HepG2 cells, OAA appeared to act as a signaling molecule able to exert a cytotoxic effect via inhibition of glycolysis by enhancing oxidative phosphorylation (OXPHOS), as well as suppression of the Akt-1/HIF pathway following apoptosis induction [52]. Altogether, data strongly support the hypothesis that AEO treatment could positively affect GBM patients. A clinical trial with AEO in GBM patients (NCT04450160) will be completed in September 2022. 

GBM cancer metabolism could also be affected by direct inhibition of the HIF-1α transcription factor. It was reported that among other biological effects exerted in cancer cells, MBM-02 (4-hydroxy-2,2,6,6-tetramethylpiperidine-N-oxyl; Tempol) could also inhibit HIF-1/2 activity, leading to cancer cell death [53,54]. MBM-2 is currently being tested in a phase 2 clinical trial (NCT04874506) along with SOC in newly diagnosed GBM patients. 

We recently described that novel 2-DG analogs could also be promising candidates for anti-GBM drugs [34]. Especially, the diacetylated 2-DG derivative—WP1122—can potently induce GBM cell death in U-87 and U-251 GBM cells [55]. The cytotoxic effect of WP1122 was synergistically potentiated with concomitant inhibition of histone deacetylases (HDAC) by sodium butyrate (NaBt) or sodium valproic (NaVPA) [55]. In December 2021, Moleculin Biotech received an FDA allowance to begin a phase 1 study of WP1122 to treat GBM [56]. 

Targeting specific cancer metabolism is an important strategy in current anti-GBM drug development. The described glycolysis inhibitors are small molecules that could penetrate BBB, thus eliminating one of the biggest challenges in GBM drug discovery. Furthermore, due to the significant difference in the metabolism of normal and cancer cells, we can expect that side effects could be mild. Finally, bearing in mind the manufacturing of drug substances, small molecules’ synthesis could be far cheaper and more accessible than biologics (such as recombinant proteins) or mAb, enabling its more efficient distribution among GBM patients. 

### 2.2. STAT3 Inhibitors as Drug Candidates for GBM Therapy

Numerous drug candidates currently tested in clinical trials represent protein/kinase inhibitors whose constitutive or elevated activity is reported in GBM cells. One of the most common targets is the signal transducer and activator of transcription 2 (STAT3) protein, effectively inhibited via ACT001 (phase 1, in combination with Pembrolizumab), BBI608 (phase 1 and 2), and WP1066 (phase 1) molecules. STAT3 is a member of the STAT family of cytoplasmic transcription factors activated by many cytokines, growth factor receptors, and downstream substrates [57]. It was reported that STAT3 phosphorylation directly correlated with GBM tumor grade, and more than 65% of GBM tumor samples showed constitutive STAT3 activity [58]. STAT3 represents an attractive target for cancer intervention due to its importance in tumor development and progression and its ability to interplay with many upstream pathways. 

ACT001 is a promising drug for treating GBM and was designated as an orphan drug for GBM by the FDA [59]. ACT001 is derived from the structural modification of parthenolide, a well-studied anti-inflammatory and anti-cancer agent [60]. Regardless of its promising anticancer effects, the clinical application of parthenolide is limited due to its instability in acidic and basic conditions [61]. Tong et al. [59] demonstrated that ACT001 directly binds to STAT3 and inhibits its activity. This may cause concern that ACT001 may have side effects due to its nonspecific targeting of healthy cells. Still, Ghantous et al. [62] reported that the nonspecific attack of ACT001’s thiol groups is restrained, most likely due to stereochemistry and conformational changes. Notably, previous clinical trials in patients with advanced solid tumors, including GBM, showed that ACT001 treatment was well-tolerated, and no dose-limiting toxicities have occurred, even at 600 mg BID (twice a day) [63]. Of the 19 patients with recurrent malignant gliomas, complete remission was observed in 1 patient with GBM, and stable disease lasting ≥6 months was seen in 3 patients [63]. Antitumor activity in malignant glioma patients was observed at a dose of 400 mg BID or lower [63]. The ongoing clinical trial (NCT05053880) includes phase 1b/2a with ACT001 combined with the anti-PD-1 molecule (Pembrolizumab) in patients with recurrent GBM. The estimated study completion date is November 2023. 

BBI608 (Napabucasin) is a small molecule that can bind to the STAT3 protein, limiting its cellular activity. Zaraquiey et al. [64] mapped the binding site and characterized the binding mode of NNI608 at STAT3, which resembles the effect of a D570K mutation in a linker domain known to interrupt STAT3 activity [65]. Han et al. [66] reported that BBI608 treatment significantly blocked GBM cells’ (U87 and LN229 cell lines) proliferation, migration, invasion, and sphere formation. Additionally, BBI608 arrested the cell cycle and induced apoptosis of GBM cells [66]. A recent clinical trial with BBI608 was completed in September 2021 (NCT02315534). It was an open-label, multicenter, phase 1 safety run-in and phase 2 study of BBI608 combined with TMZ in patients with recurrent or progressive GBM who had not received Bevacizumab treatment. In arm A, patients who were candidates for surgical resection received BBI608 as monotherapy before resection, followed by postoperative BBI608 administered in combination with TMZ. In arm B, patients who were not candidates for surgical resection received BBI608 administered daily (PO) in combination with TMZ. According to results reported in the Clinical Trials.gov database [67], 34 patients were included in the study, including 30 patients who were not candidates for surgery. In more than 70% of patients, STAT3 overexpression was confirmed. Unfortunately, there are no available data about patients’ survival (OS) or progression-free timeline (PFS) after BBI608 treatment.

WP1066 is the third STAT3 inhibitor currently registered in clinical trials. WP1066 is a small molecule that is an analog of caffeic acid and has been shown to inhibit STAT3 activity and induce GBM cell death [68,69]. Moreover, according to Iwamuru et al. [68], WP1066 treatment resulted in Bax protein activation, which further suppressed the expression of c-*myc*, *B-cell lymphoma-extra-large* (*Bcl-XL)*, and *Myeloid leukemia 1* (*Mcl-1*). On the other hand, systemic intraperitoneal administration of WP1066 in mice significantly (*p* < 0.001) inhibited the growth of subcutaneous malignant glioma xenografts during the 30-day follow-up period [68]. STAT3 overexpression is also one of the mechanisms known to correlate with cancer cells’ radio-resistance [70]. Ott et al. [71] showed that WP1066 sensitizes GBM cancer cells to radiotherapy via modulating the interaction between dendritic cells and T-cells in the tumor microenvironment. WP1066 appeared as a promising candidate for GBM or other malignant brain tumor drugs based on promising preclinical data. A phase 1 clinical trial (NCT01904123) dedicated to patients with recurrent GBM or progressive metastatic melanoma in the brain has just been completed with eight enrolled patients. The second phase 1 clinical trial WP1066 (NCT04334863) is still active for a pediatric group of patients with any progressive or recurrent brain tumor. We expect that positive results from the phase 1 study will encourage further development of WP1066 or other STAT3 inhibitors as monotherapy or as an adjuvant for radiotherapy in GBM patients. 

### 2.3. PD-1 Targeting Molecules as Drug Candidates for GBM Therapy

Programmed cell death protein 1 (PD-1) inhibits immune responses and promotes self-tolerance by modulating T-cell activity. Programmed cell death ligand 1 (PD-L1) is a transmembrane protein that supports PD-1 inhibitory action. PD-L1 combines with PD-1 to reduce the proliferation of PD-1 positive cells, inhibit cytokine secretion, and induce apoptosis. It was shown that PD-L1 promotes various malignancy development by attenuating the host immune response and tumor immune escape [72]. It was shown that PD-L1 expression on the GBM cells’ surface promotes PD-1 receptor activation in microglia, resulting in the negative regulation of T-cell responses [73]. Furthermore, GBM cells induce PD-L1 secretion by activating various receptors, such as Toll-like (TLRs), Epidermal Growth Factor (EGFR), and interferon alpha/gamma receptors (IFNα/γR). Activation of PD-1 via PD-L1 inhibits T-cell proliferation and downregulation of lymphocyte cytotoxic activity [73]. Hao et al. [74] reported that high PD-L1 expression in GBM patients is associated with poor survival, and PD-L1 may act as a prognostic predictor in GBM. On the other hand, Xue et al. [75] found that the expression of PD-L1 in glioma correlates with WHO grading and could be considered a tumor biomarker. PD-1 signaling represents a valuable therapeutic target for novel and effective drug candidates in GBM therapy with this background. 

Currently, three drug candidates are targeting the PD-1/PD-L1 pathway in GBM therapy development: Nivolumab (NCT04195139), Durvalumab (NCT02336165), and Cemiplimab (NCT03491683). All of them are monoclonal antibodies targeting the PD-1 receptor. Nivolumab is already approved for unresectable or metastatic melanoma, metastatic non-small cell lung carcinoma (NSCLC) after platinum-based chemotherapy, and metastatic renal cell carcinoma in the second-line setting [76]. Durvalumab has been approved to treat adult patients with unresectable stage III NSCLC and as a first-line treatment for adult patients with extensive-stage small-cell lung cancer (ES-SCLC) [77]. Cemiplimab is allowed as a first-line treatment in patients with NSCLC with >50% PD-L1 expression and cutaneous squamous cell carcinoma (CSCC) that is metastatic or locally advanced and not amenable to surgery. The mode of action of PD-1-targeting mAb is to activate the immune system to fight cancer. A clinical trial with Nivolumab (NCT04195139) is dedicated to newly diagnosed elderly (>65 years) patients with GBM (grade IV glioma including gliosarcoma) following surgery. The study aims to evaluate whether the combination of adjuvant Nivolumab with SOC treatment (TMZ + RT) improves overall survival outcomes for this patient population. The following active clinical trial (phase 2; NCT03367715) is a single-arm, open-label trial of Nivolumab combined with Ipilimumab and short-course radiotherapy in adult patients with newly diagnosed MGMT unmethylated GBM. NCT02017717 is a phase 3 active trial of Nivolumab combined with Bevacizumab and Ipilimumab in recurrent GBM patients. The NCT02617589 phase 3 trial of Nivolumab with RT and TMZ has also been recently completed. However, according to the data posted, there were no significant improvements in PFS and OS between Nivolumab + RT and TMZ + RT groups; thus, the study protocol or drugs’ combination has to be modified to identify the effective therapeutic window.

A clinical trial with Cemipilimab involves a combined treatment with two synthetic DNA plasmids encoding human Telomerase reverse transcriptase (hTERT), Wilms tumor protein 1 (WT-1), Prostate-specific membrane antigen (PSMA), and Interleukin 12 (IL-12), delivered by electroporation (NCT03491683). Similar to Nivolumab trials, the study is active but has not recruited any patients. The only available results regard a phase 2 study with Durvalumab and RT (in newly diagnosed GBM patients) or Bevacizumab (in recurrent GBM patients) (NCT02336165). According to published data, the overall survival rate at 12 months (OS-12) in newly diagnosed patients receiving Dervalumab and RT was estimated at 60%, whereas in recurrent GBM patients treated with Dervalumab alone or in combination with Bevacizumab, the OS did not exceed 33 months [78]. On the other hand, the assessed PFS parameter was 19.9 weeks in newly diagnosed GBM patients, whereas in recurrent GBM subjects treated with Dervalumab and Bevacizumab, the OS did not exceed 16 weeks. Unfortunately, in all patient groups, serious adverse events were reported in more than 50% of subjects, including cardiac arrest, colitis, nausea, diarrhea, convulsion, etc. Conclusions concerning Dervalumab efficacy can only be made after complex statistical analysis, and thus currently, we cannot prejudge its further development in GBM therapy. 

Currently, anti-PD-1/PD-L1 drugs do not show satisfactory efficacy in GBM patients. Available data from large-scale clinical trials with anti-PD-1 combination therapies are limited, and the published results did not demonstrate survival benefits. However, some subgroups from large-scale studies had significant benefits, suggesting the consideration of precise drug combinations [79]. Summing up, anti-PD-1/PD-L1 drug candidates for GBM have multiple challenges, and more in-depth studies are still needed.

### 2.4. DNA-Targeting Drug Candidates for GBM Therapy

DNA damage comprises a root cause of cancer development but, on the other hand, continues to provide an important avenue for chemo- and radiotherapy. Since the beginning of cancer therapy, genotoxic agents that trigger DNA damage or inhibit DNA replication and repair have been applied to stop the growth and trigger the apoptotic death of cancer cells [80]. Among the tested drug candidates for GBM therapy, a new molecule targeting DNA replication is Berubicin (NCT04915404, NCT04762069). Berubicin is a doxorubicin analog that, contrary to currently used anthracyclines, crosses the BBB and can reach brain tissue [81]. Similar to other anthracyclines, the primary mechanism of action of Berubicin involves the drug’s ability to intercalate within DNA base pairs, causing the breakage of DNA strands and inhibition of nucleic acids’ synthesis. Berubicin inhibits the enzyme topoisomerase II (TopoII), causing DNA damage and apoptosis induction [82]. Berubicin demonstrated significantly more cytotoxic activity in an orthotopic GBM murine model than TMZ [81]. Berubicin has already been tested in a phase 1 clinical trial at the MD Anderson Cancer Institute in patients with recurrent glioma [83]. According to available results, Berubicin appeared to be well-tolerated, with myelosuppression (neutropenia and thrombocytopenia) as the dose-limiting toxicity. Of the 25 treated patients, there was one complete and one partial response. Ten patients were reported with stable disease. The overall clinical efficacy rate of Berucicin treatment was 48% [83]. In 2020, the FDA approved Berubicin as an orphan drug for GBM and granted its Fast Track designation. Berubicin is currently ongoing a phase 2 clinical trial (NCT04762069) that is expected to be completed in February 2025. 

AR-67 (7-t-butyldimethylsilyl-10-hydroxycamptothecin, DB-67) is a novel camptothecins analog that interacts with topoisomerase I, a nuclear enzyme responsible for relieving DNA torsional stress and inducing cell death by stabilizing the DNA–enzyme–drug complex [84]. It was confirmed that AR-67 is characterized by improved safety and lipophilicity compared to current drugs in this class, such as Topotecan and Irinotecan [84]. A phase 1 clinical trial of AR-67 was conducted on cancer patients with solid tumors. Tang et al. [84] reported that a phase 1 study was completed to develop a population pharmacokinetic model for further drug development (NCT00389480). In 2010, a phase 2 clinical trial of AR-67 in adults patients with recurrent GBM or gliosarcoma (NCT1124539) was registered. Kumthekar et al. [85] reported that among AR-67-treated recurrent GBM patients, the drug appeared to be tolerated well and exhibited a safety profile consistent with or better than currently approved camptothecins. In patients not treated with Bevacizumab, 6/30 patients achieved the primary endpoint of 6-PFS. On the other hand, in patients who failed Bevacizumab treatment, 2/13 of patients achieved a 2-month PFS endpoint. A partial response (PR) was the best overall response in 3/45 treated patients, and stable disease (SD) was the best overall response in 7/45 patients [85].

DNA double-strand breaks induction is a mode of action of Fuzuloparib (previously Fluzoparib) that inhibits Poly-[ADP-ribose] polymerase 1 (PARP1) enzyme, leading to inhibition of DNA repair pathways, cell cycle arrest, inhibition of proliferation, and finally, cell death [86]. Fuzuloparib received its first approval in December 2020 in China to treat platinum-sensitive recurrent ovarian cancer patients after a second-line or above chemotherapy treatment [86]. Moreover, numerous phase 2 and 3 trials are currently investigating Fuzuloparib to treat other solid cancers, including cancers of the pancreas, breast, prostate, and lungs [86]. In September 2020, a phase 2 trial with Fuzuloparib combined with TMZ in recurrent GBM was registered (NCT04552977). The study is expected to be completed in August 2022. 

### 2.5. Histone Deacetylases Inhibitors in Clinical GBM Trials

Histone acetylation alternations are also associated with GBM cancer development [87]. Histone acetylation is generally associated with chromatin relaxation via neutralization of the positive charge of lysine residues [88]. On the other hand, deacetylation leads to condensed heterochromatin formation with a suppressed transcription process. Several studies confirmed that histone deacetylases (HDAC) levels are significantly upregulated in various cancer types, leading to dysregulation of the whole proteome level [89]. Altered HDAC functions have also been confirmed in GBM, leading to the upregulation of receptor tyrosine kinase (RTK)/Ras/phosphoinositide-3-kinase (PI3K), p53, retinoblastoma (Rb), EGFR, and phosphatase and tensin homolog (PTEN) signaling pathways [90]. Several studies demonstrated that HDAC blockade could inhibit tumor growth and induce the apoptosis of cancer cells, whereas normal tissue is not particularly affected. Thus, considerable interest in treating GBM using HDACi has been evoked. Several HDACi, such as Vorinostat, were reported to cross the BBB and exhibit potent cytotoxicity in monotherapy or in combination with other cytotoxic drugs or RT [91].

Vorinostat is a synthetic hydroxamic acid derivative able to bind to the catalytic domain of the HDACs [92]. This interaction results in zinc ions chelation located in the catalytic center of HDAC. As a result, the deacetylation process is inhibited, and hyperacetylated histones and transcription factors accumulate. Furthermore, in response to hyperacetylation of histone proteins, cyclin-dependent kinase p21 is activated, leading to G1 cell cycle arrest. On the other hand, hyperacetylation of non-histone proteins such as tumor suppressor p53, alpha-tubulin, and heat-shock protein 90 (HSP90) exerts an additional cytostatic response and apoptosis induction. The anticancer activity of Vorinostat resulted in its FDA approval for the treatment of cutaneous manifestations of cutaneous T-cell lymphoma (CTCL) in patients with a progressive, persistent, or recurrent disease during or following two systemic therapies [93]. 

Importantly, Vorinostat was shown to cross the blood–brain barrier [94], which is why it is considered a drug candidate in GBM therapy. The first clinical trials of Vorinostat in GBM patients showed modest single-agent activity in patients with recurrent GBM [95]. More importantly, Vorinostat appeared as a well-tolerated drug that did not cause serious side effects. This is why it is considered a synergistic compound for alkylating agents, cell cycle inhibitors, and radiation therapy. In 2018, Peters et al. [96] described the results of a phase I/II trial of Vorinostat, Bevacizumab, and daily Temozolomide treatment for recurrent malignant gliomas. Such combination therapy was well-tolerated and safe but, unfortunately, did not significantly improve the 6-month progression-free survival endpoint beyond historical controls [96]. A 2013 phase 1 trial of Vorinostat, in combination with Bevacizumab and Irinotecan in recurrent GBM patients, was completed; however, the results obtained have not been posted in the Clinical Trials Database. Additionally, our recent preclinical studies support the hypothesis about the beneficial effects of HDACi as synergistic agents for anti-GBM treatment. Using the acetylated 2-DG analog—WP1122—combined with HDACi (sodium butyrate, NaBt, and sodium valproate, NaVPA), we obtained a strong cytotoxic effect against GBM cells [55]. Thus, the combination of previously described glycolysis inhibitors (DCA and AEO) in more advanced stages of their clinical development with Vorinostat could be an attractive therapeutic option for GBM treatment. 

A recently registered clinical trial with Vorinostat included a combination treatment with Pembrolizumab (anti-PD-1 mAb), TMZ, and RT for newly diagnosed GBM patients (NCT03426891). The study is active and is planned to be completed this year.

### 2.6. Other Current Drug Candidates for GBM Therapy

As summarized in Table 1, there are other drug candidates for GBM therapy with unique mechanisms of action. New compounds, such as WEE1 tyrosine kinase inhibitor AZD1775, CXC chemokine receptor type 4 (CXCR4) inhibitor USL311, or multi-kinase inhibitor AEE788, are already registered drugs, such as Metformin and Tamoxifen, that are approved for other indications. The majority of the protocols proposed include combined treatment with GBM SOC or other available chemotherapeutics. Unfortunately, most drug candidates are still in the early development phases. Moreover, to become an alternative for TMZ first-line drug, the drug candidates must first show significant clinical efficacy in recurrent patients, who are usually heavily treated and in an advanced stage of tumor development, often with multidrug resistance. Some hopes are related to drugs granted Orphan Drug status and the Fast Track designation pathway, which may be introduced to the clinic soon.

## 3. Perspectives

Despite aggressive treatment with the current standard of care (maximal surgical resection with adjuvant radiotherapy and TMZ), the 5-year survival for GBM remains at a dismal <2% and has not changed significantly for decades. Figure 1 summarizes the timeline that clearly illustrates the impasse in GBM therapy development. Almost 30 years after the first chemotherapeutics like Carmustine, in 2005 TMZ was included in SOC GBM protocol. A few years later, Bevacizumab was registered, but as discussed earlier, its clinical efficacy is limited and does not improve the OS of GBM patients. Developing chemotherapy strategies include systemic (oral or intravenous) therapies and local, intra-tumoral delivery (such as Convection Enhanced Delivery, CED) of chemical compounds, immunotherapeutics, vaccines, or biologics. New generations of drug candidates often target cancer-specific proteins. However, disrupting cell division, DNA repair of the cellular metabolism is still considered an anticancer tool. Several drug candidates are considered for potential GBM treatment. However, the most promising are dichloroacetate (DCA) (glycolysis inhibitor), ACT001 (STAT3 inhibitor), and Berubicin (TopoII poison), granted with Fast Track designation and an Orphan Drug status, which supports their faster transition into clinics (Figure 1). DCA is currently in a phase 2 clinical trial, whereas ACT001 and Berubicin have entered phase 1. Unfortunately, bearing in mind the early stages of their clinical development, their implementation in therapy will also be time-consuming, and it is difficult to predict when the upgrade of SOC will be possible.

Nevertheless, GBM chemotherapy is still a current challenge that requires the implementation of new, highly effective drug combinations to achieve significant improvement. Given the steadily increasing number of patients diagnosed with GBM, the hope for the future is a breakthrough that could significantly extend patients’ lives.

## Figures and Tables

**Figure 1 biomedicines-10-01001-f001:**
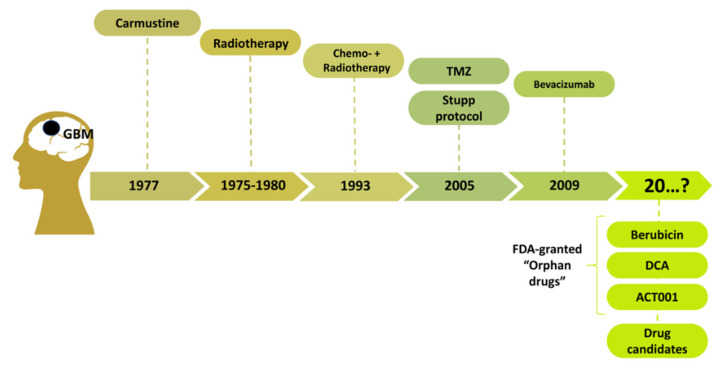
The timeline of GBM therapy up to current SOC. The most advanced and promising drug candidates for GBM therapy with FDA-granted ‘Orphan Drug’ status are included.

**Table 1 biomedicines-10-01001-t001:** Summary of clinical trials with drug candidates for GBM chemotherapy (2016–2021).

**Drug Candidate/** **Administration Route** **(IV—Intravenous, PO—Oral, IM—** **Intramuscular Injection)**	**Mechanism of Action**	**Indication**	**Clinical Phase**	**Clinical Trial ID (Status)**
**Chemotherapy**				
AZD1775 -PO	selective inhibitor of Wee1 tyrosine kinase	-recurrent GBM	1	NCT02207010 ^C^
Chlorogenic acid -IV	immunomodulatory action	-advanced GBM	2	NCT03758014
USL311 -IV	C-X-C motif chemokine receptor 4 (CXCR4) inhibitor	-solid tumors (alone) -recurrent GBM (with Lomustin)	1 2	NCT02765165 *
ACT001 -IV	nuclear factor kappa B (NF-kB) and signal transducer and activator of transcription 3 (STAT3) inhibitor; immunomodulatory action	-recurrent GBM (with Perbrolizumab)	1	NCT05053880 ^R^
Prostate-specific membrane antigen (PSMA ADC) -IV	PSMA and tumor neovascularization inhibitor	-GBM -gliosarcoma	2	NCT01856933 ^C^
Cabazitaxel -IV	cell death induction by microtubule stabilization	-TMZ refractory GBM	2	NCT01866449 ^C^
Dichloroacetate (DCA) -PO	glycolysis inhibitor	-recurrent GBM	2A	NCT05120284 ^A^
-t-butyldimethylsiltyl-10-hydroxy-camptothecin (AR-67) -IV	DNA replication inhibition	-recurrent GBM	2	NCT01124539
Acalabrutinib (ACP-196) -PO	irreversible Bruton’s tyrosine kinase (BTK) inhibitor	-recurrent GBM	1/2	NCT02586857 ^A^
VB-111 -IV	neovascularization inhibitor	-GBM -recurrent GBM (prior to and after surgery)	2	NCT04406272 ^R^
-Trans sodium crocetinate -IV	enhancement of the oxygenation of hypoxic tissues	-recurrent GBM (with TMZ and RT)	3	NCT03393000 *
Dacomitinib (PF-00299804) -PO	pan-ErbB inhibitor	-primary GBM (prior surgery) -recurrent GBM with EGFR amplification or mutation	2 2	NCT01112527 ^C^ NCT01520870 ^C^
Durvalumab -IV	PD-1 receptor inhibitor	-GBM -recurrent GBM (after Bevacizumab)	2	NCT02336165 ^C^
Infigratinib (BGJ398) -PO	fibroblast growth factor receptor (FGFR) tyrosine kinase inhibitor	-primary GBM (patients who are not candidates for surgery)	2	NCT01975701 ^C^
BBI608 (Napabucasin) -PO	STAT3 inhibitor	-primary GBM (monotherapy prior resection and with TMZ postoperative) -recurrent GBM (with TMZ, without Bevacizumab treatment)	1 2	NCT02315534 ^C^
AEE788 -PO	multi-targeted kinase inhibitor with potent inhibitory activity against the ErbB and VEGF receptor family of tyrosine kinases	-recurrent GBM	1/2	NCT00116376 ^C^
Fuzuloparib (Fluzoparib) -PO	Poly [ADP-ribose] polymerase 1/*2* (PARP1/2) inhibitor	-recurrent GBM (with TMZ)	2	NCT04552977 ^A^
Sorafenib -PO	multi-kinase inhibitor facilitating apoptosis, mitigating angiogenesis, and suppressing cell proliferation	-recurrent GBM (with TMZ)	2	NCT00597493 ^C^
Tamoxifen -PO	selective estrogen receptor (ER) modulator, mitochondrial complex I inhibitor	-recurrent GBM	2	NCT04765098 ^A^
Berubicin -IV	Topoisomerase II (Topo II) stabilization, DNA replication inhibitor	-recurrent GBM	1/2	NCT04915404 ^A^ NCT04762069 ^R^
WP1066 -PO	STAT3 inhibitor	-recurrent GBM -metastatic melanoma in the brain -pediatric patients with any progressive or recurrent malignant brain tumor	1 1	NCT01904123 ^C^ NCT04334863 ^R^
BAL101553 (Lisavanbulin) -IV	promotes tumor cell death by modulating the spindle assembly checkpoint	-advanced, recurrent solid tumors or GBM	1/2	NCT02895360 ^C^
Nivolumab -IV	PD-1 receptor inhibition	-recurrent GBM (with Bevacizumab and Ipilimumab)	3	NCT02017717 ^A^
**Adjuvant for radiotherapy**				
Mibefrabil -PO	calcium channels (T-type and Orai1-3) blocker	-recurrent GBM	1	NCT02202993 ^C^
Axitinib -PO	VEGFR1-3, c-KIT, and platelet-derived growth factor (PDGFR) inhibitor	-GBM	2	NCT01508117 *
Nivolumab -IV	PD-1 receptor inhibitor	-MGMT-unmethylated GBM (with Ipilimumab and RT) -GBM (newly diagnosed elderly patients) (with/without TMZ and RT) -MGMT-unmethylated GBM (with TMZ and RT)	2 2 3	NCT03367715 ^A^ NCT04195139 ^A^ NCT02617589 ^C^
Durvalumab -IV	PD-1 receptor inhibitor	-newly diagnosed GBM with unmethylated MGMT (with RT)	2	NCT02336165 ^C^
Ipilimumab -IV	cytotoxic T cell antigen 4 (CTLA-4) inhibitor, immunomodulatory action	-MGMT-unmethylated GBM (with TMZ and RT)	2	NCT03367715
2-hydroxyoleic acid (2-OHOA) -PO	cell cycle arrest, autophagy induction, changes in membrane–lipid composition	-newly diagnosed GBM	1	NCT03867123 ^A^
INO-5401 -IM followed by electroporation (EP)	synthetic DNA plasmids encoding for hTERT, WT-1 and PSMA	-newly diagnosed GBM (with TMZ and RT +INO-9012)	1	NCT03491683 ^A^
INO-9012 -IM followed by electroporation (EP)	synthetic DNA plasmid encoding Interleukin 12 (IL-12)	-newly diagnosed GBM (with TMZ and RT + INO5401)	2	NCT03491683 ^A^
Cemiplimab -IM followed by electroporation (EP)	PD-1 receptor inhibitor (mAb)	-newly diagnosed GBM (with TMZ and RT)	2	NCT03491683 ^A^
Arginine pegylated with 2000-molecular-weight polyethylene glycol (ADI-PEG20) -IM	arginine deprivation agent	-newly diagnosed GBM (with TMZ and RT)	1	NCT04587830 ^R^
MBM-02 (Tempol) -PO	hypoxia-inducing factor 1/2 (HIF-1/2) inhibitor	-newly diagnosed GBM (with TMZ and RT)	2	NCT04874506 ^A^
Anhydrous enol-oxaloacetate (AEO) -PO	glycolysis inhibitor	-primary GBM (after surgery along with TMZ and RT)	2	NCT04450160 ^A^
Depatuxizumab Mafodotin (ABT414) -IV	epidermal-growth-factor-receptor (EGFR)-targeting antibody–drug conjugate consisting of the mAb 806 and a toxic payload, monomethyl auristatin F	-newly diagnosed GBM (with TMZ and RT)	1	NCT01800695 ^C^
Vorinostat -IV	histone deacetylases (HDAC) inhibitor	-newly diagnosed GBM (with TMZ and RT +/− Pembrolizumab)	1	NCT03426891 ^A^
Metformin -PO	5′AMP-activated protein kinase (AMPK) inhibitor (metabolism, angiogenesis, inflammation, and cancer stem cells control), and proliferation inhibitor (via insulinemia and glycemia reduction)	-newly diagnosed GBM (with TMZ and RT)	2	NCT02780024 ^A^
Onfekafusp alfa -IV	immunoglobulin, anti-(human fibronectin ed-b domain) (synthetic human clone L19 scfv fragment fusion protein with human tumor necrosis factor alpha (TNF), trimer	-newly diagnosed GBM (with TMZ and RT)	1/2	NCT04443010 ^A^
Everolimus -PO	selective mammalian target of rapamycin (mTOR) kinase inhibitor	-newly diagnosed GBM (with TMZ and RT and Bevacizumab)	2	NCT00805961 ^C^
Bavituximab -IV	monoclonal antibody directed against anionic phospholipids with potential antineoplastic activity	-newly diagnosed GBM (with TMZ and RT)	2	NCT03139916 ^A^
Gliadel -IV	DNA and RNA alkylating agent	-newly diagnosed GBM (with TMZ and RT and Bevacizumab)	2	NCT01186406 ^TT^

^C^—Completed. *—Terminated (business reasons not related to safety). ^R^—Recruiting. ^A^—Active, not recruiting. ^TT^—Terminated due to toxicity.

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
