# Peer review of "Looking for the Holy Grail—Drug Candidates for Glioblastoma Multiforme Chemotherapy"

_biomedicines, 2022, doi:10.3390/biomedicines10051001_

Round 1

Reviewer 1 Report

This manuscript describes the results of clinical trials for the most recent candidate compounds used for chemotherapy of patients diagnosed with Glioblastoma multiforme.

This review may help generate new ideas for the researchers in the development of new compounds or new drug combinations that are effective in prolonging the life or treatment of patients diagnosed with GBM.

This manuscript can be accepted for publication if the following aspects can be modified:

Line 28: The first-line standard of care (SOC) treatment is based on complete surgical resection, followed by concomitant radio- and chemotherapy (concomitant administration of…)

Line 80: with an overall survival of approximately 30 weeks [18].

Line 94: Currently, Carmustine is used to treat certain brain tumors, including GBM, brain-stem glioma, medulloblastoma, astrocytoma, ependymoma, and metastatic brain tumors.

Hard to read text, please consider rephrasing the sentence.

Line 114: Carmustine (as an implantation in a resection cavity or intravenous administration) treatment showed prolonged OS.

Line 171: To overcome this, To overcome this issue/problem,

Line 175: It is postulated that aerobic glycolysis is advantageous to tumor growth as it increases cell biomass, generating glucose-6-phosphate for the pentose phosphate pathway and, in turn, providing precursors for fatty acid, amino acid, and nucleic acid synthesis [30], which may maintain glycolysis as the primary pathway for glucose metabolism, even in the presence of oxygen.

Hard to read text, please consider rephrasing the sentence.

Line 183: Normally activated in periods of starvation, (unclear, please rephrase)

Line 189: While not all types of cancer show high dependence on anaerobic glucose metabolism, pancreatic ductal adenocarcinoma (PDAC), and 190 glioblastoma multiforme (GBM), some of the most aggressive and lethal types of tumors 191 are entirely dependent on glycolysis [34].

Hard to read text, please consider rephrasing the sentence.

Line 296: anticancer agent à anti-cancer agent

Line 327: Unfortunately, there are no available data showing the clinical efficacy of BBI608 with regard to patients’ survival or progression-free timelines.

This phrase may be wordy, consider changing the wording (concerning patients` survival..)

Line 367: Currently, there are three drug candidates targeting…

    Sounds better:

Currently, three-drug candidates are targeting the PD-1/PD-L1 pathway in 367 GBM therapy development: …

Line 385: The following active clinical trial (2 phase; NCT03367715) ….. 2 phases, plural form!

Line 466: expected to be completed inàon August 2022.

     On … it seems that preposition use may be incorrect here.

Line 557: Developing chemotherapy strategies are multidirectional, including systemic therapies or local, intra-tumoral delivery of chemical compounds and immunotherapeutics, vaccines, biologics, and others

Hard to read text, please consider rephrasing the sentence.

Pay attention to punctuation in the table.

Overall, I suggest a minor revision of this work.

Author Response

Reviewer 1

This manuscript describes the results of clinical trials for the most recent candidate compounds used for chemotherapy of patients diagnosed with Glioblastoma multiforme.

This review may help generate new ideas for the researchers in the development of new compounds or new drug combinations that are effective in prolonging the life or treatment of patients diagnosed with GBM.

  • Thank you for the positive comment and appreciation of our efforts. We corrected our manuscript according to Your suggestions. Please, find below our responses to Your comments.

This manuscript can be accepted for publication if the following aspects can be modified:

Line 28: The first-line standard of care (SOC) treatment is based on complete surgical resection, followed by concomitant radio- and chemotherapy (concomitant administration of…)

  • The sentence has been corrected.

Line 80: with an overall survival of approximately 30 weeks [18].

  • The sentence has been corrected.

Line 94: Currently, Carmustine is used to treat certain brain tumors, including GBM, brain-stem glioma, medulloblastoma, astrocytoma, ependymoma, and metastatic brain tumors.

Hard to read text, please consider rephrasing the sentence.

  • The sentence has been corrected: “Currently, Carmustine is used in primary (GBM, medulloblastoma, astrocytoma, ependymoma) and metastatic brain tumors therapy.”

Line 114: Carmustine (as an implantation in a resection cavity or intravenous administration) treatment showed prolonged OS.

  • The sentence has been corrected.

Line 171: To overcome this, To overcome this issue/problem,

  • The sentence has been corrected.

Line 175: It is postulated that aerobic glycolysis is advantageous to tumor growth as it increases cell biomass, generating glucose-6-phosphate for the pentose phosphate pathway and, in turn, providing precursors for fatty acid, amino acid, and nucleic acid synthesis [30], which may maintain glycolysis as the primary pathway for glucose metabolism, even in the presence of oxygen.

Hard to read text, please consider rephrasing the sentence.

  • The sentence has been corrected: “It is postulated that aerobic glycolysis is advantageous to tumor growth as it increases cell biomass. It was shown that several signaling pathways implicated in cell proliferation also regulate metabolic pathways that incorporate nutrients into biomass. Thus, glycolysis could be maintained as the primary pathway for glucose metabolism, even in the presence of oxygen [30].”

Line 183: Normally activated in periods of starvation, (unclear, please rephrase)

  • The sentence has been corrected to: “HIF-1a regulates the transcription of glucose transporters and glycolytic enzymes genes and augments mitochondrial respiration [32]”.

Line 189: While not all types of cancer show high dependence on anaerobic glucose metabolism, pancreatic ductal adenocarcinoma (PDAC), and 190 glioblastoma multiforme (GBM), some of the most aggressive and lethal types of tumors 191 are entirely dependent on glycolysis [34].

Hard to read text, please consider rephrasing the sentence.

  • The sentence has been corrected to: “While not all types of cancer rely on anaerobic glucose metabolism, the most aggressive types of tumors, such as pancreatic ductal adenocarcinoma (PDAC), and glioblastoma multiforme (GBM), entirely depend on glycolysis [34].”

Line 296: anticancer agent à anti-cancer agent

  • The sentence has been corrected.

Line 327: Unfortunately, there are no available data showing the clinical efficacy of BBI608 with regard to patients’ survival or progression-free timelines.

This phrase may be wordy, consider changing the wording (concerning patients` survival..)

  • The sentence has been corrected to: “Unfortunately, there are no available data about patients' survival (OS) or progression-free timeline (PFS) after BBI608 treatment.”

Line 367: Currently, there are three drug candidates targeting…

    Sounds better:

Currently, three-drug candidates are targeting the PD-1/PD-L1 pathway in 367 GBM therapy development: …

  • The sentence has been corrected.

Line 385: The following active clinical trial (2 phase; NCT03367715) ….. 2 phases, plural form!

  • The sentence has been corrected from 2 phase to phase 2 to make it clear. “The following active clinical trial (phase 2; NCT03367715) is a single-arm………..

Line 466: expected to be completed inàon August 2022.

     On … it seems that preposition use may be incorrect here.

  • The manuscript has been corrected by MDPI English Editing Service. “The study is expected to be completed in August 2022.”

Line 557: Developing chemotherapy strategies are multidirectional, including systemic therapies or local, intra-tumoral delivery of chemical compounds and immunotherapeutics, vaccines, biologics, and others

Hard to read text, please consider rephrasing the sentence.

  • The sentence has been corrected to: “Developing chemotherapy strategies include systemic (oral or intravenous) therapies and local, intra-tumoral delivery (such as Convection-enhanced delivery, CED) of chemical compounds, immunotherapeutics, vaccines, or biologics.”

Pay attention to punctuation in the table.

  • The Table has been checked and corrected.

Overall, I suggest a minor revision of this work.

  • Thank you for Your recommendation.

Reviewer 2 Report

The authors summarized current and potential future novel therapies for glioblastoma. The manuscript is generally well-prepared, and the therapeutic strategies were well-introduced. Only minor suggestions for the revision

1) Line 88: what is ‘PI’ group. Did authors mean BI group?

2) Figure 1 does not look not very useful for the overall manuscript. I suggest adding drug targeting types next to the orphan drugs. For example: DCA:metabolism drug

Author Response

Reviewer 2

The authors summarized current and potential future novel therapies for glioblastoma. The manuscript is generally well-prepared, and the therapeutic strategies were well-introduced. Only minor suggestions for the revision

  • Thank you for the positive comment and appreciation of our efforts. We corrected our manuscript according to Your suggestions. Please, find below our responses to Your comments.

1) Line 88: what is ‘PI’ group. Did authors mean BI group?

- The line has been corrected. It should be BI group.

2) Figure 1 does not look not very useful for the overall manuscript. I suggest adding drug targeting types next to the orphan drugs. For example: DCA:metabolism drug

- Figure 1 discussion has been included into the Perspective section. The requested explanation of the drug types has been explained in the text along with the Figure 1 description.

Reviewer 3 Report

It was a review paper about some of the current therapeutic agents used for glioblastoma treatment during the clinical tests. Here are some comments on this study that should be considered before publication:

  • Please introduce all the abbreviations at the first-time usage.
  • "In 1997, the FDA approved Gliadel wafers to treat recurrent GBM and treat newly diagnosed high-grade glioma (World Health Organization (WHO) grade III and IV) in 2003” please rewrite this sentence. It is not clear.
  • There are some grammatical mistakes in the text that should be corrected.
  • “Numerous pre- 218 clinical studies also suggest positive effects of DCA coadministered with compounds cur- 219 rently used to treat other diseases but showing anticancer properties in several cancer 220 models [43].” Please add more references fort his sentence.
  • Please improve the quality of perspective section.
  • Please add some figure data to the text.

Author Response

Reviewer 3

It was a review paper about some of the current therapeutic agents used for glioblastoma treatment during the clinical tests.

  • Thank you for the comments and fruitful feedback. We corrected our manuscript according to Your suggestions. Please, find below our responses to Your comments.

Here are some comments on this study that should be considered before publication:

  • Please introduce all the abbreviations at the first-time usage.
  • We fully agree with the Reviewer. The manuscript has been carefully checked and all missing abbreviations (such as MDR/Pgp line 75; radiotherapy (RT) line 139; ATP line 213; HIF-1a line 223; STAT3 line 351; Bcl-XL line 399; Mcl-1 line 399; EGFR line 432; hTERT, WT-1, PSMA and IL-12 line 470-471; CXCR4 line 608) has been introduced at the first-time usage.
  • "In 1997, the FDA approved Gliadel wafers to treat recurrent GBM and treat newly diagnosed high-grade glioma (World Health Organization (WHO) grade III and IV) in 2003” please rewrite this sentence. It is not clear.

- The sentence has been corrected to: “In 1997, the FDA approved Gliadel wafers for recurrent GBM therapy. A few years later, in 2003, Gladel was accepted for primary high-grade glioma (grade III and IV according to the World Health Organization (WHO) classification) treatment.”

  • There are some grammatical mistakes in the text that should be corrected.
  • The manuscript has been corrected by MDPI English Editing service.
  • “Numerous pre- 218 clinical studies also suggest positive effects of DCA coadministered with compounds cur- 219 rently used to treat other diseases but showing anticancer properties in several cancer 220 models [43].” Please add more references fort his sentence.
  • The section has been updated according to the Reviewer’s suggestion and the following citations has been added.
  1. Skeberdytė, A.; Sarapinienė, I.; Aleksander-Krasko, J.; Stankevičius, V.; Sužiedėlis, K.; Jarmalaitė, S. Dichloroacetate and salinomycin exert a synergistic cytotoxic effect in colorectal cancer cell lines. Sci Rep 2018, 8(1), 17744. 
  2. Verma, A.; Lam, Y.M.; Leung, Y.C., Hu, X.; Chen, X.; Cheung, E.; Tam, K.Y. Combined use of arginase and dichloroacetate exhibits anti‐proliferative effects in triple negative breast cancer cells. J Pharm Pharmacol 2019, 71(3), 306–315. 
  3. Li, B.; Li, X.; Xiong, H.; Zhou, P.; Ni, Z.; Yang, T.; Zhang, Y.; Zeng, Y.; He, J.; Yang, F.; Zhang, N.; Wang, Y.; Zheng, Y.; He, F. Inhibition of COX2 enhances the chemosensitivity of dichloroacetate in cervical cancer cells. Oncotarget 2017, 8(31), 51748–51757. 
  4. Lucido, C.; Miskimins, W.; Vermeer, P. Propranolol promotes glucose dependence and synergizes with dichloroacetate for anti-cancer activity in HNSCC. Cancers 2018, 10(12), 476. 
  5. Prokhorova, I.V.; Pyaskovskaya, O.N.; Kolesnik, D.L.; Solyanik, G.I. Influence of metformin, sodium dichloroacetate and their combination on the hematological and biochemical blood parameters of rats with gliomas C6. Exp Oncol 2018, 40(3), 205–210.
  6. Kolesnik, D.L.; Pyaskovskaya, O.N.; Yakshibaeva, Y.R.; Solyanik, G.I. Time-dependent cytotoxicity of dichloroacetate and metformin against Lewis lung carcinoma. Exp Oncol 2019, 41(1), 14–19.

“Numerous preclinical studies also suggest positive effects of DCA coadministered with compounds currently used to treat other diseases but showing anticancer properties in several cancer models [43]. Skeberdytė et al. [44] found that administration of DCA and the antibiotic salinomycin exerted a synergistic cytotoxic effect by inhibiting the expression of proteins related to multidrug resistance in colorectal cancer cell lines. On the other hand, a combined administration of recombinant arginase and DCA had antiproliferative effects in triple-negative breast cancer due to the activation of p53 and cell cycle arrest [45]. The subsequent study performed in cervical cancer cells showed that COX2 inhibition by celecoxib makes cervical cancer cells sensitive to DCA treatment [46]. In the head and neck cancer model, the combination of DCA with propranolol, also appeared to be an efficient cell death inducer [47]. A positive effect of DCA coadministration with metformin, a drug widely used to treat diabetes, was demonstrated in a preclinical model of glioma [48] as well as in a low metastatic variant of Lewis lung carcinoma (LLC) [49].”

  • Please improve the quality of perspective section.
  • The Perspectivesection has been corrected. Also, Figure 1 has been moved and discussed in this section to correlate discussed perspectives in the context of GBM therapy development. 
  • Please add some figure data to the text.
  • Figure description has been added to the text.

Round 2

Reviewer 3 Report

Thanks for addressing the comments.